# Umbilical Cord, Maternal Milk, and Breastfed Infant Levetiracetam Concentrations Monitoring at Delivery and during Early Postpartum Period

**DOI:** 10.3390/pharmaceutics13030398

**Published:** 2021-03-17

**Authors:** Ivana Kacirova, Milan Grundmann, Hana Brozmanova

**Affiliations:** 1Department of Clinical Pharmacology, Faculty of Medicine, University of Ostrava, 70300 Ostrava, Czech Republic; ivana.kacirova@osu.cz (I.K.); hana.brozmanova@osu.cz (H.B.); 2Department of Clinical Pharmacology, Department of Laboratory Medicine, University Hospital Ostrava, 70852 Ostrava, Czech Republic

**Keywords:** levetiracetam, delivery, breastfeeding

## Abstract

(1) To obtain objective information about levetiracetam transplacental passage and its transport into colostrum, mature milk, and breastfed infants, we analyzed data from women treated for epilepsy between October 2006 and January 2021; (2) in this cohort study, maternal, umbilical cord, milk, and infant serum concentrations were measured at delivery, 2–4 days postpartum (colostrum) and 7–31 days postpartum (mature milk). Paired umbilical cord serum, maternal serum, breastfed infant serum, and milk concentrations were used to assess the ratios of umbilical cord/maternal serum, milk/maternal serum, and infant/maternal serum concentrations. The influence of combined treatment with enzyme-inducing antiseizure medication carbamazepine was assessed; (3) the umbilical cord/maternal serum concentration ratio ranged between 0.75 and 1.78 (mean 1.10 ± 0.33), paired maternal and umbilical cord serum concentrations were not significantly different, and a highly significant correlation was found between both concentrations. The mean milk/maternal serum concentration ratio was 1.14 ± 0.27 (2–4 days postpartum) and 1.04 ± 0.24 (7–31 days postpartum) while the mean infant/maternal serum concentration ratio was markedly lower (0.19 ± 0.13 and 0.14 ± 0.05, respectively); (4) levetiracetam was found in the umbilical cord at a concentration similar to those in maternal serum. All of the breastfed infant serum concentrations were below the reference range used for the general epileptic population.

## 1. Introduction

The majority of women with epilepsy need to follow antiseizure medication (ASM) treatment during pregnancy and breastfeeding. A significant shift from “old” to “new” generation of ASM, especially lamotrigine (LTG) and levetiracetam (LEV), was described worldwide over the past decade [1,2]. Changes in LEV clearance during pregnancy were observed in some studies with a significant decline in plasma concentrations during pregnancy compared to the baseline levels before or after pregnancy [3,4,5,6]. The concentration of LEV increases rapidly after delivery, and baseline pre-pregnancy levels are reached within the first weeks after pregnancy. However, the effects of pregnancy on LEV disposition vary considerably between individuals and monitoring of LEV levels during pregnancy and after delivery is recommended [5,6]. An extensive transfer of LEV from mother to fetus and into breast milk was reported by some authors in small study groups or case-reports [4,5,6,7,8,9,10,11,12,13,14,15,16,17]. Despite this, breastfed infants had measured insignificant serum concentrations, and LEV is usually considered compatible with breastfeeding [4,7,8,9,10,11,12,13,14,15,16,17]. However, the infant should be monitored for drowsiness, adequate weight gain, and developmental milestones, especially in younger, exclusively breastfed infants, and when using combinations of anticonvulsants [18]. Comedication with enzyme-inducing ASM (carbamazepine, phenobarbital, phenytoin) enhances the metabolism of LEV, resulting in a 20–30% reduction in plasma LEV concentrations [19]. Nevertheless, information on the influence of ASM interactions on LEV concentrations during delivery and breastfeeding are negligible. Because of the incomplete knowledge in LEV treatment of breastfeeding women, we followed up not only LEV transplacental passage but also its transport into breast milk and suckling infants over more than 14-years period. Because breast milk in the first 3–5 days postpartum primarily contains colostrum and by the end of the first week the milk is mature, LEV concentrations were analyzed in both colostrum and mature milk and compared those with the maternal and infant serum concentrations.

The aim of our study is to determine transplacental passage of LEV and its transport to colostrum, mature maternal milk, and breastfed infants. The influence of co-medication with enzyme-inducing antiseizure drug carbamazepine was also analyzed.

## 2. Materials and Methods

Inclusion criteria: Information of epileptic women treated by LEV at delivery and/or during breastfeeding were analyzed. Request forms for routine therapeutic drug monitoring (TDM) and maternal serum, umbilical cord serum, milk, and breastfed infant serum concentrations collected in our department between October 2006 and January 2021 were used as the data source. The study was appropriately reviewed and approved by the local Ethics Committee (Reference number 487/2020, The Ethics Committee of University Hospital Ostrava, Czech Republic). Written consent before enrollment to the study was not undersigned with regard to routine TDM. Exclusion criteria: serum and/or milk samples and request forms for routine TDM from other patients which did not fulfil inclusion criteria.

The “delivery” subgroup collected data from 14 pregnant women and their infants (7 girls, 2 boys, and 5 were not stated). Maternal and umbilical cord serum concentrations were collected simultaneously at birth.

In the “colostrum” subgroup we evaluated the information from 58 women and 54 breastfed infants (1 pair of twins; 33 girls, 20 boys, 1 was not stated). Colostrum, maternal and breastfed infant serum samples were drawn between the 2nd and 4th postnatal day (median 3 days), samples were taken in the morning before the first maternal LEV dose. For statistical analysis, we received both maternal, colostrum and infant samples from 43 patients (including 1 pair of twins), only maternal and colostrum samples from 8 patients, only maternal and infant serum samples from 5 patients; and five times we analyzed only the breastfed infant serum samples.

The “milk” subgroup comprised data from 8 women and 10 suckling infants (5 girls, 5 boys) in which milk and serum samples were obtained from the 7th to 31th day after delivery (median 12 days) mostly in the morning before the first maternal LEV dose. All three samples (maternal, milk and infant) were taken from 2 patients, only maternal and milk samples were observed from 7 women, only maternal and infant serum sample from 1 patient, and seven times we analyzed only breastfed infant serum samples. Serum samples from four breastfed infants were obtained during all three periods (delivery–colostrum–mature milk). For LEV concentration analysis, there was taken one vial of colostrum or milk with volume of about 4.0 mL.

Total serum and milk concentrations were measured by liquid chromatography with UV/VIS detection method. Analysis was provided from milk samples without first defatting. 50 μL of internal standard (α-methyl-5,5 dimethyl-2oxo-1-pyrolidin acetamid) and 50 μL of serum or milk were extracted to 1 mL of dichloromethane in the presence of 20 μL 1 mol/L NaOH. Dichloromethane layer was evaporated to dryness and residue was dissolved in mobile phase composed of water:methanol:acetonitril:triethylamin (85:10:5:0.002) at pH 5.2. GraceSmart RP 18 column 150 × 2.1 mm was used for analysis and detection was performed with spectrometric detector at 205 nm. Performance characteristics of the method were as follows: linearity was found between 0.5 and 100 mg/L, both for serum and milk. The accuracy and precision were validated by U.S. Food and Drug Administration rules; the within-day and between-day precision and accuracy were studied at three concentration levels in both matrices. At tested concentrations, recovery in serum was between 96.5 and 102.5, the coefficient of variations was 1.4–6.5%, recovery in milk was 98.8–111.1% and the coefficient of variations was 5.2–8.1%, respectively. The lower limit of quantification (LLoQ) was estimated as 0.9 mg/L. The method was quality controlled in external quality control Instand (Germany) twice a year. For statistical calculations half of the LLoQ was used for infants with concentrations less than the LLoQ [17,20].

Apparent oral clearance (Cl) was calculated for LEV: Cl (L/kg) = daily dose (mg/kg)/maternal serum concentration (mg/L) [21]. Paired umbilical cord serum, maternal serum, breastfed infant serum, and milk concentrations of LEV were used for the assessment of the ratio of umbilical cord/ maternal serum, milk/maternal serum and infant/maternal serum concentrations. In the “colostrum” subgroup the influence of combination with carbamazepine (CBZ), enhancing the metabolism of LEV was followed. We also evaluated the relationship between LEV umbilical cord serum, maternal serum, infant serum, and milk concentrations.

Statistical analysis was performed using GraphPad Prism version 5.00 for Windows, GraphPad Software (San Diego, CA, USA; www.graphpad.com; accessed on 1 February 2021). The D’Agostino and Pearson omnibus normality test was applied for test if the values come from a Gaussian distribution. Thereafter, we used the paired *t*-test (when the values follow the Gaussian distribution) or the nonparametric Wilcoxon signed rank test for the comparison of two matched groups, and the Pearson correlation test (the Gaussian distribution) or the Spearman nonparametric correlation test was used for the correlation analysis. A value of *p* < 0.05 was considered statistically significant.

## 3. Results

Basic characteristics of mothers and their infants is presented in Table 1.

At delivery, the LEV concentrations varied from 1.3 to 41.5 mg/L in the maternal serum and between 2.3 and 36.6 mg/L in the umbilical cord serum (Table 2). A highly significant correlation was observed between the umbilical cord serum concentrations and the maternal serum concentrations (*p* < 0.0001, the Pearson correlation coefficient = 0.9118, Figure 1). The umbilical cord/maternal serum concentration ratio ranged between 0.75 and 1.78 and paired maternal and umbilical cord serum concentrations were not significantly different (*p* = 0.6159, the paired *t*-test), see Table 2. Fifty percent of the maternal serum concentrations were found in the reference range 12–46 mg/L [19], and 50% were lower. As well, 50% of the umbilical cord serum concentrations were measured in the reference range used for the general epileptic population, and 50% were lower (none below the LLoQ). LEV monotherapy was prescribed to 50% of women, 43% used combination with lamotrigine or topiramate, and 1 woman used a triple combination with lamotrigine and valproic acid.

In the colostrum subgroup (2–4 days after delivery), LEV concentrations varied from 1.4 to 35.7 mg/L in the maternal serum, from 1.7 to 27.3 mg/L in the milk, and from 0.5 to 6.3 mg/L in the breastfed infant serum (Table 3). A highly significant correlation was observed between milk and maternal serum concentrations (*p* < 0.0001, the Spearman correlation coefficient = 0.9101), infant serum and milk concentrations (*p* = 0.001, the Spearman correlation coefficient = 0.4788), and also, between infant and maternal serum concentrations (*p* < 0.0001, the Spearman correlation coefficient = 0.5535). The mean milk/maternal serum concentration ratio was 1.14 ± 0.27, and milk concentrations were found to be slightly but significantly higher than paired maternal serum concentrations (median 8.9 versus 7.7 mg/L, *p* = 0.0063, the Wilcoxon signed rank test). The mean of the infant/maternal serum concentration ratio (0.19 ± 0.13) has been estimated to be markedly lower as well as the mean of the infant serum/milk concentration ratio (0.18 ± 0.12). Comparison of the distribution of the milk/maternal serum concentration ratio and the infant/maternal serum concentration ratio is shown in Figure 2. Twenty nine percent of the maternal serum concentrations were analyzed in the reference range and 71% were lower, as well as in the milk. None of the infant serum concentrations were analyzed in the reference range used for the general epileptic population, and 16 infants (30%) exhibited values less than the LLoQ. LEV monotherapy was prescribed in 46% of women, 51% used bi-combination, and 3% (two women) triple-combination with other ASM. Statistical analysis for evaluation of drug interaction was not performed due to a small number of patients using combination with carbamazepine. However, an effect of CBZ on the LEV maternal apparent oral clearance is evident and this co-medication markedly decreased the LEV concentrations in breastfed infants in which every concentrations reached values less than the LLoQ (Table 3).

In the later date after delivery (7–31 postpartum days), the maternal serum concentrations varied from 4.4 to 28.6 mg/L, the milk concentrations from 2.8 to 21.6 mg/L, and the infant serum concentrations from 0.5 to 5.1 mg/L (Table 4). The mean milk/maternal serum concentration ratio was 1.04 ± 0.24, and milk concentrations were not found to be significantly different than the paired maternal serum concentrations (median 10.1 versus 10.1 mg/L, *p* = 0.4688, the Wilcoxon signed rank test). The mean of the infant/maternal serum concentration ratio (0.14 ± 0.05) has been estimated to be markedly lower as well as the mean of the infant serum/milk concentration ratio (0.14 ± 0.06). A significant correlation was found between the milk and maternal serum concentrations (*p* = 0.0123, the Spearman correlation coefficient = 0.8829), and paired milk and maternal serum concentrations were not significantly different (*p* = 0.4688, the Wilcoxon signed rank test). Fifty percent of the maternal serum concentrations were measured in the reference range and 50% were lower similarly to the milk (43% of samples was measured in the reference range used for the general epileptic population and 57% was lower). By contrast, none of the infant serum concentrations was analyzed in the reference range used for the general epileptic population, one concentration was lower than the LLoQ. LEV monotherapy was prescribed in 40% of women and 60% used bi-combination with other ASM. LEV serum concentrations obtained from four breastfed infants in all three time points (delivery–colostrum–mature milk) are shown in Figure 3.

Given the colostrum (2–4 days after delivery) and the later period (7–31 days after delivery) subgroups together, paired milk concentrations were slightly but significantly higher than maternal serum concentrations (median 9.0 mg/L versus 7.8 mg/L, *p* = 0.0043, the Wilcoxon signed rank test) and highly significant correlation between milk and maternal serum concentrations was observed (*p* < 0.0001, the Spearman correlation coefficient = 0.9110), see Figure 4. The mean milk/maternal serum concentrations ratio was close to one (mean 1.13 ± 0.26) in comparison with the mean infant/maternal serum concentrations ratio which reached approximately a sixth of the value (mean 0.19 ± 0.13) similarly as the mean of infant serum/milk concentrations ratio (mean 0.17 ± 0.11), see Table 4.

## 4. Discussion

In our study, especially the number of simultaneously analyzed maternal serum, milk and infant serum concentrations were greater (from one center using a consistent methodology) than in all previous studies using varying and incomparable criteria [4,7,8,9,10,11,12,13,14,15,16,17], see Table 5.

The mean umbilical cord/maternal serum concentration ratio was 1.10, and paired maternal and umbilical cord serum concentrations were not significantly different. So, we confirmed the free transplacental passage of LEV [4,7,8,10,11,12]. As new, a highly significant correlation was found between the umbilical cord serum and the maternal serum concentrations.

During breastfeeding, we observed a similar result of the milk/maternal serum concentration ratio both in colostrum and mature milk as in the previous reports of Tomson et al., Johannessen et al., and Dinavitser et al. [4,7,8,9]. We did not confirm the considerable accumulation of LEV in breast milk reported by Kramer et al. [13]. However, milk concentrations have been found to be slightly but significantly higher than paired maternal serum concentrations especially in the “colostrum” subgroup (2–4 days postpartum). The mean milk/maternal serum concentration ratio was close to one compared to the mean infant/maternal serum concentration ratio, which reached approximately a sixth of the value. The result is supported by the comparison of the distribution of these two paired ratios in Figure 2, in which the moving of the infant/maternal serum concentration ratio to the lower values (i.e., “to the left”) is apparent. The mean and median, respectively of infant/maternal serum concentration ratio we observed was higher than in the studies by Tomson et al. [4] and Birnbaum et al. [17] in the “colostrum” subgroup (2–4 days postpartum) and in the study of Birnbaum et al. [17] in the “milk” subgroup (7–31 days postpartum). Compared to Birnbaum et al.’s study [17], we collected venous blood samples (not dried blood spots) from breastfed infants who were younger with a median age of 3 days (range 2–31 days) versus a median age of 13 weeks (range 6–20 weeks). Moreover, we analyzed both colostrum and milk concentrations simultaneously compared to Birnbaum et al.’s study [17] in which milk concentrations were not measured. In the “colostrum” subgroup (2–4 days postpartum), we demonstrated a highly significant correlation between milk and maternal serum concentrations and between infant serum concentrations and both milk and maternal serum concentrations. Statistical analysis for the evaluation of drug interaction with carbamazepine was not performed due to the small number of this combination. However, an effect of CBZ to enhance the LEV maternal apparent oral clearance is evident and moreover this combination markedly decreased LEV concentrations in breastfed infants. In the “milk” subgroup (7–31 days after delivery), a significant correlation was found between the milk and maternal serum levels, and compared to “colostrum” subgroup paired milk and maternal serum levels were not significantly different. The reason of this difference can be a smaller number of patients in the “milk” subgroup.

Fifty percent of maternal serum and 43% of milk LEV concentrations were measured in the reference range used for the general epileptic population but none of the infant serum level has reached this values. Breastfed infant exposure to ASM in milk varies, depending on multiple factors: maternal serum drug concentration, the milk/maternal serum concentration ratio, the milk volume ingested by infant, and the absorption, metabolism, and excretion of the drug in the infant [22]. The elimination half-life of plasma LEV was reported 16–18 h in both nursed and formula-fed infants [4,9], which is longer than the 6–8 h found in adults [19]. Conversely, the clearance of LEV in neonates was found higher than predicted and increased significantly during the first week of life into the range seen in older children and exceeded values reported in adults. By Sharpe et al. [23], infants may have reduced the capacity for tubular reabsorption of LEV and altered hydrolysis to the main acid metabolite UCB L057. Both elimination pathways increase in function during the first week of life [23]. Moreover, the volume of distribution in neonates in a study of Merhar et al. was 0.89 L/kg compared to 0.6–0.7 L/kg in children and 0.5–0.7 L/kg in adults. Because neonates have a higher total body water content than children and adults, a larger volume of distribution is expected because LEV is water-soluble and therefore has a distribution that reflects total body water [24,25]. According to Nicolas et al. [26] it could be tentatively hypothesized that the immature intestinal villi in children, with a lower surface/volume ratio and a reduced absorptive surface, may influence the LEV absorption. Although the mechanisms have not been studied in detail, it seems to be probable that the decline in LEV levels in breastfed infants during the early postpartum period is caused by a combination of the increased renal and non-renal elimination, the higher volume of distribution and the lower intestinal absorption [23,24,26].

The number of mothers receiving LEV during lactation was greater in our study (from one center using the same methodology) than in all reported studies using varied and incomparable criteria. Moreover, we analyzed a larger number of infant serum concentrations than in all previous studies put together [7,10,11,12,14,16,17,18,19,20]. From clinical point of view, in the case of LEV the degree of its exposure in breastfed infants was markedly smaller than during gestation. Likewise, although milk concentrations have been found to be slightly higher or similar than paired maternal serum concentrations (especially in the “colostrum” subgroup), infant serum concentrations reached approximately only a sixth of the maternal value. Therapeutic monitoring of LEV serum concentrations in breastfed infants is not mandatory; however, if signs of potential adverse reactions are noted, infant serum concentrations should be analyzed.

There are some limitations in our study. Total LEV concentrations were analyzed, but it is only 3% bound to plasma proteins [19] and it is not necessary to analyze its free fraction. The patients on concomitant therapy with carbamazepine with a higher potential to enhance the clearance of LEV were not analyzed due to the small number of this combination. However, a trend for carbamazepine’s effect is evident. The volume of ingested breast milk, the timing of feedings, and information on exclusively breastfeeding in the included infants are not known; however, this study presented a direct measure of LEV concentrations in venous blood samples of breastfed infants along with the measurements of drug concentrations in both mothers, colostrum and mature milk. We were not able to demonstrate any relation between LEV concentrations and clinical effects in either the fetus or in breastfed infants. No attempt was made to correlate LEV concentrations to maternal seizure control during pregnancy or in the lactation period. However, we hope that these new data from our study may be important for treating epilepsy during pregnancy and breastfeeding and for knowledge of fetuses’ and breastfed infants’ exposure to LEV.

## 5. Conclusions

This study systematically analyzed the transplacental passage and transport of levetiracetam to colostrum, mature milk, and breastfed infants in the largest group of patients ever reported to our knowledge. LEV has been found in umbilical cord serum at concentrations similar to those in maternal serum and the highly significant correlation between umbilical cord and maternal serum concentrations was observed. The mean milk/maternal serum concentration ratio was close to one compared to the mean infant/maternal serum concentration ratio, which reached approximately only a sixth of the value. Moreover, none of the infant serum concentrations was analyzed in the reference range used for the general epileptic population. Therefore, routine monitoring of infant serum LEV concentration is not required. However, observation of breastfed infant is advisable, and if signs of potentially adverse reactions are noted, infant serum concentration should be measured.

## Figures and Tables

**Figure 1 pharmaceutics-13-00398-f001:**
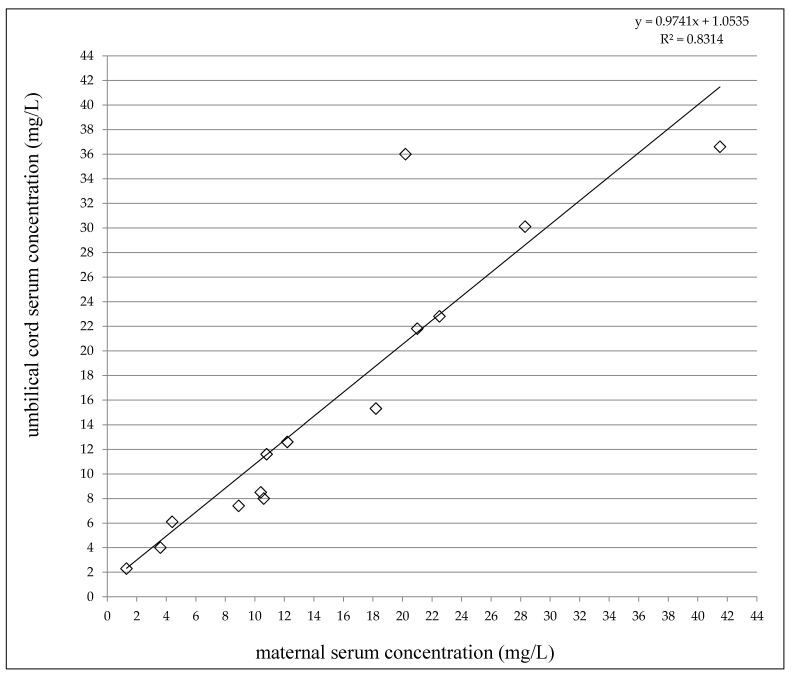
Correlation between umbilical cord serum and maternal serum concentrations of levetiracetam. Number = 14, *p* < 0.0001, the Pearson correlation coefficient = 0.9118.

**Figure 2 pharmaceutics-13-00398-f002:**
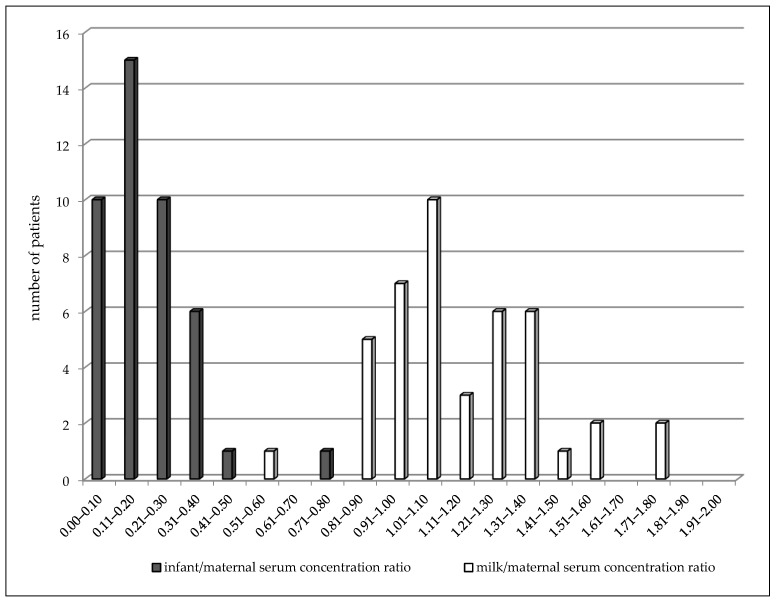
Comparison of distribution of paired milk/maternal serum concentration ratio (median = 1.07) and infant/maternal serum concentration ratio (median = 0.16) of levetiracetam 2–4 days after delivery. Number = 43, *p* < 0.0001 (Wilcoxon signed rank test).

**Figure 3 pharmaceutics-13-00398-f003:**
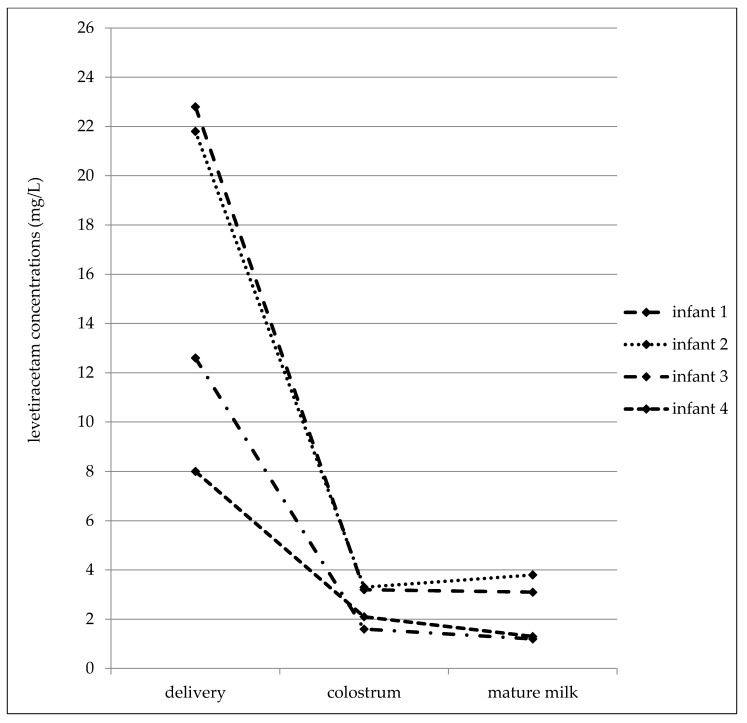
Levetiracetam serum concentrations obtained from four breastfed infants in all three time points (delivery–colostrum–mature milk).

**Figure 4 pharmaceutics-13-00398-f004:**
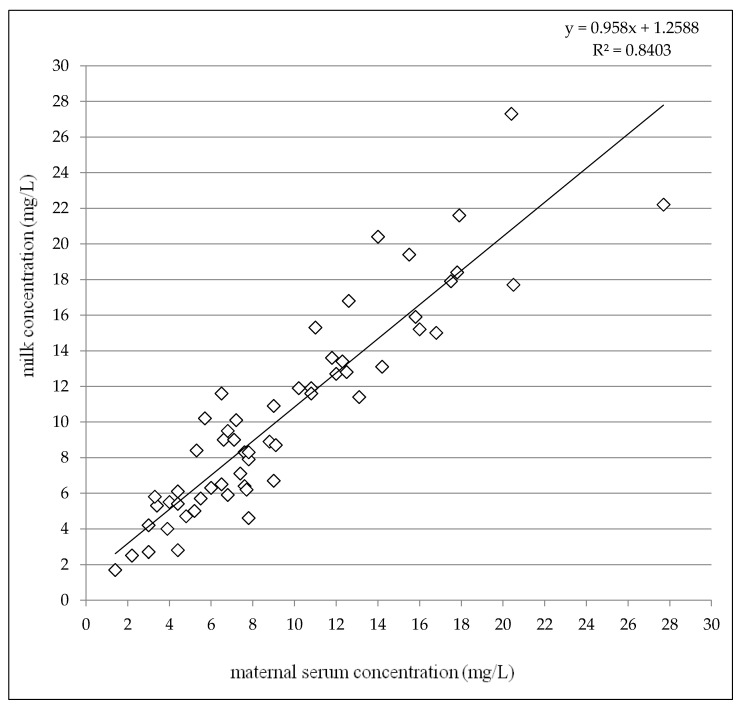
Correlation between milk concentrations and maternal serum concentrations of levetiracetam 2–31 days after delivery altogether. N = 58, *p* < 0.0001, the Spearman correlation coefficient = 0.9110.

**Table 1 pharmaceutics-13-00398-t001:** Basic characteristics of mother and their infants.

Mothers/Infants	Characteristics		Delivery		2–4 Days after Delivery		7–31 Days after Delivery
		N	mean ± SD (range)	N	mean ± SD (range)	N	mean ± SD (range)
Mothers:	age (years)	14	27 ± 5 (18–37)	58	29 ± 5 (18–41)	8	30 ± 3 (27–34)
Concomitant antiseizure medication	lamotrigine (mg/day)	6	300 ± 155 (200–600)	21	371 ± 112 (150–550)	6	275 ± 96 (200–400)
	valproic acid (mg/day)	1	500	2	875 ± 530 (500–1250)	0	-
	topiramate (mg/day)	1	100	3	183 ± 76 (100–250)	0	-
	carbamazepine (mg/day)	0	-	6	717 ± 184 (400–900)	1	400
	zonisamide (mg/day)	0	-	2	525 ± 35 (500–550)	2	525 ± 35 (500–550)
	lacosamide (mg/day)	0	-	1	200	0	-
Infants:	weight (kg)	10	3.3 ± 0.4 (2.6–3.9)	51	3.2 ± 0.4 (2.0–4.0)	10	3.1 ± 0.5 (2.1–3.7)
	length (cm)	10	49 ± 2 (45–50)	51	48 ± 3 (37–52)	10	48 ± 3 (42–50)
	female	7		33		5	
	male	2		20		5	

**Table 2 pharmaceutics-13-00398-t002:** Dosage, apparent oral clearance (Cl), maternal (M), and umbilical cord (UC) serum concentrations of levetiracetam and its ratio at delivery.

Levetiracetam	Weight (kg)	Dose mg/Day)	Dose/kg (mg/kg)	Cl (L/kg)	M (mg/L)	UC (mg/L)	UC/M Ratio
number	11	12	11	11	14	14	14
median range	75 68–114	1500 250–3000	17.0 3.7–34.5	1.39 0.83–3.37	11.5 1.3–41.5	12.1 2.3–36.6	1.04 0.75–1.78
mean ± SD	81 ± 15	1438 ± 813	18.6 ± 10.5	1.48 ± 0.69	* 15.3 ± 10.9	* 15.9 ± 11.6	1.10 ± 0.33

* *p* = 0.6159 (the paired *t*-test)-maternal serum versus umbilical cord serum concentrations.

**Table 3 pharmaceutics-13-00398-t003:** Dosage, maternal apparent oral clearance (Cl), maternal serum (M), milk (Mi), and infant (I) serum levetiracetam concentrations and their ratio 2–4 days after delivery in monotherapy (and/or combination with “neutral” drugs) versus combination with carbamazepine (CBZ).

LEV Mono + Neutral Drugs	Weight (kg)	Dose (mg/day)	Dose (mg/kg)	Cl (L/kg)	M (mg/L)	Mi (mg/L)	I (mg/L)	Mi/M Ratio	I/M Ratio	I/Mi Ratio
number	45	50	44	44	51	47	49	47	45	41
median range	73 55–125	1500 500–4000	17.6 6.8–50.0	2.22 1.23–5.38	8.8 1.4–35.7	9.0 1.7–27.3	1.4 0.5–6.3	1.09 0.59–1.79	0.16 0.03–0.71	0.17 0.03–0.52
mean ± SD	76 ± 14	1625 ± 870	21.4 ± 12.1	2.47 ± 1.03	10.1 ± 6.5	10.5 ± 5.7	1.8 ± 1.4	1.16 ± 0.27	0.20 ± 0.13	0.18 ± 0.12
LEV + CBZ										
number	5	5	5	5	5	4	5	4	4	3
median range	68 57–76	1000 500–2500	17.2 6.6–33.8	3.65 0.97–4.57	4.8 2.2–7.4	5.3 2.5–7.1	0.5 0.5–0.5	0.97 0.87–1.14	0.12 0.07–0.23	0.11 0.07–0.20
mean ± SD	67 ± 9	1100 ± 822	16.5 ± 11.0	3.37 ± 1.43	5.0 ± 2.1	5.1 ± 2.0	0.5 ± 0.0	0.99 ± 0.11	0.13 ± 0.07	0.13 ± 0.07
Total										
number	50	55	49	49	56	51	54	51	49	44
median range	72 55–125	1500 500–4000	17.2 6.6–50.0	2.33 0.97–5.38	* 7.7 1.4–35.7	* 8.9 1.7–27.3	1.2 0.5–6.3	1.07 0.59–1.79	0.16 0.03–0.71	0.17 0.03–0.52
mean ± SD	75 ± 14	1577 ± 872	20.9 ± 11.9	2.56 ± 1.10	9.6 ± 6.4	10.1 ± 5.6	1.7 ± 1.4	1.14 ± 0.27	0.19 ± 0.13	0.18 ± 0.12

* *p* = 0.0063 (the Wilcoxon signed rank test)-maternal serum versus milk concentrations.

**Table 4 pharmaceutics-13-00398-t004:** Dosage, maternal apparent oral clearance (Cl), maternal (M) serum, milk (Mi), and infant (I) serum levetiracetam concentrations and their ratio 7–31 days after delivery and 2–31 days after delivery altogether.

	Weight (kg)	Dose (mg/Day)	Dose/kg (mg/kg)	Cl (L/kg)	M (mg/L)	Mi (mg/L)	I (mg/L)	Mi/M Ratio	I/M Ratio	I/Mi Ratio
7–31 days after delivery										
number	6	7	6	6	8	7	10	7	3	2
median range	70 57–73	2000 1000–3000	34.1 13.9–52.6	3.48 1.78–7.31	* 10.1 4.4–28.6	* 10.1 2.8–21.6	2.0 0.5–5.1	1.06 0.64–1.40	0.12 0.11–0.20	0.14 0.10–0.18
mean ± SD	68 ± 6	2214 ± 699	34.1 ± 13.5	4.14 ± 2.30	12.9 ± 8.0	11.3 ± 6.0	2.3 ± 1.4	1.04 ± 0.24	0.14 ± 0.05	0.14 ± 0.06
Total 2–31 days after delivery										
number	56	62	55	55	64	58	64	58	52	46
median range	71 55–125	1500 500–4000	17.9 6.6–52.6	2.33 0.97–7.31	** 7.8 1.4–35.7	** 9.0 1.7–27.3	1.4 0.5–6.3	1.07 0.59–1.79	0.16 0.03–0.71	0.17 0.03–0.52
mean ± SD	74 ± 14	1649 ± 873	22.3 ± 12.7	2.73 ± 1.34	10.0 ± 6.7	10.2 ± 5.7	1.8 ± 1.4	1.13 ± 0.26	0.19 ± 0.13	0.17 ± 0.11

* *p* = 0.4688 (the Wilcoxon signed rank test)-maternal serum versus milk concentrations. ** *p* = 0.0043 (the Wilcoxon signed rank test)-maternal serum versus milk concentrations.

**Table 5 pharmaceutics-13-00398-t005:** Review of literature (Ref-reference, N-number, M-maternal, UC-umbilical cord, Mi-milk and I-infant concentrations of levetiracetam) [4,7,8,9,10,11,12,13,14,15,16,17].

Ref	N		Dose (mg/Day)	M (mg/L)	UC (mg/L)	UC/M Ratio			
[4]	13		1000–3000	1.9–20.4	1.2–31.3	mean 1.15 range 0.56–2.00			
[10]	10		750–5000		mean 22.75 ± 21.81 range 4.00–76.00	mean 1.12 ± 0.46 range 0.57–2.17			
[12]	5		2000–3000	3.5–24.9	5.7–29.6	mean 1.21 range 0.92–1.62			
[7]	4		2000–3500	11.4–48.3	16.5–54.8	mean 1.14 range 0.97–1.45			
[8]	4					mean 0.87 range 0.69–1.03			
[11]	1		2000	17.0	23.0	1.35			
	N	Postpartum time	Dose (mg/day)	M (mg/L)	Mi (mg/L)	Mi/M ratio	N	I (mg/L)	I/M ratio
[17]	58	6–20 weeks	500–5500	median 24.0 range 0.3–73.5			58	median 0.9 range 0.9–4.5	median 0.05 range 0.02–0.20
[9]	14	3–22 weeks	mean 2517			mean 0.88 range 0.23–1.10			
[14]	12	4 days 2–3 months			lower than maternal levels				
[4]	11	4–23 days	1000–3000	mean 15.4 range 4.6–34.2	mean 16.0 range 5.8–35.7	mean 1.05 range 0.78–1.55	10	mean 1.9 range 0.7–3.4	mean 0.13 range 0.07–0.22
[7]	7 1–5	3–5 days 2 we-10 mon first 8 weeks	1500–3500	mean 13.8 range 4.8–29.8	mean 12.5 range 4.8–26.0	mean 1.00 range 0.76–1.33 range 0.93–1.22	7 1	<1.7–2.5 2.5–2.9	
[8]	4	3–4 days 2 w-4 mo				mean 0.93 range 0.76–1.04 range 0.93–1.32	4	below limit of quantification	
[15]	1		2250 3000	18.8 21.0	16.0–33.6 29.0–51.7				
[11]	1	5 days infant day 8	2000		<3.0		1	<3.0	
[13]	1	1–2 weeks		5.4	16.8	3.09	1	1.0 (96 h after stop of lactation)	
[16]	1	10 days	3000				1	2.1	

## Data Availability

Authors declare that take full responsibility for the data, the analyses and interpretation, and the conduct of the research; that they have full access to all of the data; and that they have the right to publish all data. Authors were not participations in industry-sponsored research and corporate activities for evaluation of a manuscript.

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
