# Peer review of "Umbilical Cord, Maternal Milk, and Breastfed Infant Levetiracetam Concentrations Monitoring at Delivery and during Early Postpartum Period"

_pharmaceutics, 2021, doi:10.3390/pharmaceutics13030398_

Round 1

Reviewer 1 Report

Thank you for the opportunity to review this manuscript.

Please clearly state the aims of the investigation in the Introduction.

The second paragraph in Materials and Methods is repeated.

Is the LC analysis method based on a previously-published method?

More information is required regarding colostrum and milk collection and analysis.

What volumes of colostrum or milk were collected?

Were whole colostrum and milk analysed or were the samples defatted first?

What volume of colostrum or milk was used for analysis?

Has the analytical method been validated previously for milk samples?

Is there a difference in the concentrations of levetiracetam between samples collected before a breastfeed or expression (lower cream content) compared to after a breastfeed or expression (higher cream content)?

Table 1

I am very concerned about the weights of the infants 7-31 days. The heaviest infant would have been on the 11th centile on the World Health Organisation weight-for-age chart, and the lightest less than the 1st centile.

Table 3

How were 47 milk samples obtained from 8 women? Serial collections?

Table 4 is mis-labelled as Table 1.

Infant intestinal absorption of levetiracetam should be discussed e.g. Nicolas et al Biopharm. Drug Dispos. 38: 209–230 (2017)

It is interesting that the infant serum concentrations of levetiracetam are more-or-less constant from colostrum to mature milk stages (Figure 3) given the large increase in volume of milk ingested, and therefore dose of levetiracetam, between the two stages. Although the volume of ingested breastmilk was not measured, some reference to literature values should be made, and then discussion as to why a greater intake of levetiracetam does not result in a higher serum concentration in the infants.

Reviewer 2 Report

The authors of the paper “Umbilical cord, maternal, milk, and breastfed infant levetiracetam concentrations monitoring at delivery and during early postpartum period”  analysed the transplacental passage and transport of levetiracetam to colostrum, mature milk and breastfed infants. Research design is appropriate, and methods are adequately described. The result might be more clearly presented, sometimes it is difficult to read the tables. Tables provide lots of data, which is good, but for average reader it might be difficult to analyse presented data. Some minor typing errors are present – lines 67-72 are repeated; line 232 should be Table 4 instead of Table 1.

Reviewer 3 Report

This is a high-quality manuscript devoted to an important and interesting research topic. The manuscript is well organized and well written. Figures and tables are informative, and statistical analysis is extensive. I have only several minor comments to the contents of this manuscript:

1. Lines 67 & 70 - the same paragraph is repeated twice. Please correct.

2. Line 81 - the "milk" subgroup - the collected samples represent the pre-dose (trough) concentrations? Please clarify in the methods, and consider to elaborate on the timing of collection of these samples and its influence on the results of this dataset in the Discussion. 

3. Discussion - please divide the text into several paragraphs, to improve its readability.
